# A robot intervention for adults with ADHD and insomnia–A mixed-method proof-of-concept study

Siri Jakobsson Støre[1]*, Maria Tillfors[1], Charlotte Angelhoff[2,3], Annika Norell-Clarke[4,5]

**1** Department of Social and Psychological Studies, Karlstad University, Karlstad, Sweden, **2** Crown Princess Victoria's Child and Youth Hospital, Linköping, Sweden, **3** Department of Biomedical and Clinical Sciences, Linköping University, Linköping, Sweden, **4** Faculty of Health Sciences, Kristianstad University, Kristianstad, Sweden, **5** School of Law, Psychology and Social Work, Örebro University, Örebro, Sweden

* siri.store@kau.se

## Abstract

### Objective

To investigate individual effects of a three-week sleep robot intervention in adults with ADHD and insomnia, and to explore participants' experiences with the intervention.

### Methods

A proof-of-concept study with a mixed-methods design (n = 6, female = 4) where a repeated ABA single-case study was combined with interviews. Data were collected with the Consensus Sleep Diary, wrist actigraphy, questionnaires on symptoms of insomnia, arousal, emotional distress, and ADHD, and through individual interviews.

### Results

Visual analysis of the sleep diary and actigraphy variables did not support any effects from the robot intervention. Half of participants reported clinically relevant reductions on the Insomnia Severity Index from pre- to post-intervention. No changes regarding ADHD or arousal. Thematic analysis of the interviews resulted in three themes: (1) A pleasant companion, (2) Too much/not enough, and (3) A new routine.

### Conclusion

Adjustments of the intervention ought to be made to match the needs of patients with both ADHD and insomnia before the next trial is conducted.

## Introduction

Attention-deficit/hyperactivity disorder (ADHD) is one of the most common neurodevelopmental disorders. The prevalence of *persistent* (childhood onset) ADHD in adults has been

**Data Availability Statement:** The data is publicly available from the ISRCTN registry (https://doi.org/10.1186/ISRCTN11007746).

**Funding:** The authors received no specific funding for this work.

**Competing interests:** The authors have declared that no competing interests exist.

found to be 2.58%, compared with 6.76% in *symptomatic* (childhood or later onset) ADHD in Western countries [1]. The Diagnostic and Statistical Manual of Mental Disorders (DSM-5) diagnostic criteria of ADHD include symptoms of inattention and/or hyperactivity and impulsivity [2]. These symptoms are associated with functional impairments when it comes to education, work, and/or social relationships [3]. According to the DSM-5, there are three types of presentations of ADHD: (1) combined presentation, (2) predominately inattentive presentation, and (3) predominately hyperactive-impulsive presentation. The population of adults who meet the diagnostic criteria is heterogenous in terms of combinations of core symptoms and comorbidities [4], of which sleep disorders such as insomnia are common [5, 6]. In fact, "restless sleep" was previously a diagnostic criterion of ADHD in the DSM-III [7]. Sleep disturbances have been found to be more common in the combined presentation of ADHD compared with the predominately inattentive presentation [8], whereas in other studies the opposite has been found to be true [9]. Insomnia was in one study found to be as prevalent as in more than forty percent of adult ADHD patients [10].

The DSM-5 diagnostic criteria of chronic insomnia (> 3 months) are prolonged sleep onset latency, wake after sleep onset, and/or early morning awakenings, without being able to fall back asleep, causing significant impairment or distress [2]. Initial insomnia is the most common type of insomnia in adults with ADHD [11], but this can be difficult to differentiate from other sleep disorders (e.g., circadian rhythm sleep disorder). Furthermore, as Wajszilber, Santiseban and Gruber (pp. 453–454) write, "comorbid sleep disorders are often overlooked and left untreated in ADHD populations." Thus, a thorough sleep screening is of utmost importance in both clinical and research settings [12].

ADHD and insomnia are both characterized by atypical levels of arousal. Hyperarousal has, since long, been regarded as an important mechanism in the emergence and maintenance of insomnia [13, 14]. As Owens et al. (p. 553) write, "available data indicate that underlying neurochemical and anatomical modulation of sleep, arousal and attention overlap" [15]–that is, certain brain regions are involved in all three both functionally and anatomically. It has also been suggested that ADHD, like insomnia, should be conceptualized as a 24-hour disorder [16]. Hence, it is interesting to study interventions that aim to reduce arousal and improve sleep in people with comorbid ADHD and insomnia.

A meta-analysis of studies on sleep in adults with ADHD found statistically significant differences between those with ADHD and those without ADHD regarding seven out of nine subjective sleep parameters (longer sleep onset latency, more psychosomatic symptoms during sleep onset, higher number of night awakenings, more general sleep problems, lower sleep quality, lower sleep efficiency, and more daytime sleepiness), as well as on two out of five objective (actigraphy) parameters (longer sleep onset latency and lower sleep efficiency) [17]. Most research studies on the relationship between ADHD and insomnia are correlational [16], but there is some support of a bidirectional relationship [18]. Sleep problems may mimic or intensify symptoms of ADHD through mechanisms such as emotional dysregulation and reduced executive functions [19, 20]. A recent experimental study on the effects of sleep restriction on ADHD symptoms in adolescents found that parents of adolescents with ADHD (but not the adolescents themselves) reported more inattention in the sleep restriction group compared with the control group. Perhaps more interesting, adolescents in the sleep restriction group reported *less* hyperactivity/impulsivity compared with the control group [18]. A current analysis of six genome-wide association studies (one for insomnia, and five for other psychiatric disorders including ADHD) was not in support of the causal role of insomnia in ADHD [21]. Medicating ADHD with psychostimulants, e.g., methylphenidate, can both disturb sleep as they affect neurotransmitter systems that promote wakefulness [15], and enhance sleep through beneficial effects on ADHD symptoms [22]. Taken together, the relationship

between ADHD and insomnia is complex and insufficiently researched, not the least in the adult population.

Cognitive Behavioral Therapy (CBT-I) is the gold standard treatment of insomnia [23]. There is not much evidence on CBT-I for adult ADHD patients, but Jernelöv et al. made an important contribution with their pilot study on a CBT-I group treatment adjusted for ADHD, of which the results were in support of the treatment [24]. Relaxation techniques, sometimes incorporated into CBT-I, are associated with sleep enhancement, for instance, mindfulness exercises and slow deep breathing techniques [25, 26]. A literature review on sleep interventions in adults with ADHD identified six studies: three studies on bright light therapy, Jernelöv et al. study on CBT-I [24], one study on melatonin supplements, and one study on weighted blankets [27]. One of Surman and Walsh conclusions was that understudied low-risk interventions may improve sleep in this patient group [28]. Of relevance here, treatment with weighted chain blankets (deep pressure stimulation) was found to have beneficial effects on insomnia in patients with ADHD [25]. Another sleep device produced to enhance sleep through sensory stimulation is the Somnox sleep robot [29].

The Somnox sleep robot aims to guide people into slow deep breathing and help them to fall asleep faster by simulating breath sounds and chest movement. The first randomized wait-list-controlled study on the effects of the sleep robot in adults with insomnia showed no statistically significant effects on a group level [30]. Considering the neurochemical and anatomical overlap in sleep, arousal and attention, it is possible that the robot might have greater effects in adults with comorbid ADHD and insomnia. The sleep robot might also benefit adults with ADHD as it is easy to use and does not demand the ability to stay focused throughout a Cognitive Behavioral Therapy program. The aims of the study were to assess whether a three-week sleep robot intervention had individual effects in adults with ADHD and insomnia, and how the initial results could be understood in light of participants' experiences with the robot. The research questions were: (1) Does the sleep robot intervention affect participants' symptoms of insomnia, somatic arousal, anxiety, depression, and ADHD? (2) How do participants experience the sleep robot?

## Materials and methods

### Study design

The study was a mixed-methods study with an explanatory sequential design, that is, an initial quantitative phase followed by a qualitative phase with the purpose to understand the quantitative results in depth [31]. As Creswell and Plano Clark (p. 326) have stated, "When used in combination, quantitative and qualitative methods complement each other and provide a more complete picture of the research problem" [31]. The study followed the guidelines for Good Reporting of a Mixed Methods Study (GRAMMS) [32]. The study was approved by the Swedish Ethical Review Authority (DNR 2020–06975). Written consent to study participation was obtained. The study was conducted at Karlstad University between 3 January and 11 April 2022. The study was registered in the ISRCTN Registry (ISRCTN11007746), retrospectively due to the pilot nature of the study. The authors confirm that all ongoing and related trials for this intervention are registered. We have no conflicts of interest to declare, specifically no collaboration with the company which has produced the sleep robot.

The quantitative phase of the study had a repeated ABA single-case design (SCED). SCEDs have been defined as "designs that are applied to experiments in which one entity is observed repeatedly during a certain period of time under different levels of at least one independent variable" [33, p. 56]. Among the listed types of experimental designs in mixed methods studies, Creswell and Plano Cark include the single-case experimental design [31]. They also

emphasize the importance of following the methodological requirements of the chosen experimental design.

Participants kept a daily sleep diary throughout the experiment, that is, six weeks in total: two baseline weeks, three intervention weeks, and one post-intervention week. In addition to the sleep dairy, pre-, post- and 1-month follow-up assessments were conducted with different questionnaires (see outcome measures for more information). The qualitative phase entailed an interview with the participants post treatment, focusing on their experiences with the robot and the intervention as a whole.

## Participants and procedure

Participants were recruited through the university webpage and through social media. The sample size of 6 was determined based on the feasibility nature of the study (i.e., new patient group) due to the lack of clear beneficial effects in the preceding study [30]. The screening was conducted via phone by the first author (SJS). Participants were screened for insomnia severity by *the Insomnia Severity Index* (ISI) [34], comorbid sleep disorders by *the Duke Structured Interview for Sleeping Disorders* (DSISD) [35], and psychiatric disorders by *the Mini International Neuropsychiatric Interview* (M.I.N.I.) [36]. Participants were previously diagnosed with ADHD, which was confirmed by the Adult ADHD Self-Report Scale (ASRS) [37]. Additionally, the participants' demographic characteristics were registered (gender, marital status, number of children in the household, highest level of education, employment, and whether they were born in Sweden or not). In terms of inclusion criteria, participants had to (1) speak Swedish fluently, (2) be over 18 years of age, (3) be previously diagnosed with ADHD, and (4) meet the diagnostic criteria of insomnia. Exclusion criteria were (1) meet the diagnostic criteria of another untreated sleep disorder, and (2) meet the diagnostic criteria of another current psychiatric disorder. Prescribed medical treatments of ADHD or insomnia had to be stable (same dosage for at least 2 months) prior to the study. Out of the ten persons who showed interest in the study, six completed the screening. The other four either stated that they were not interested after more information about the study (n = 2), or did not answer by phone/e-mail at the screening appointment (n = 2). See Fig 1 for the flow chart of the study.

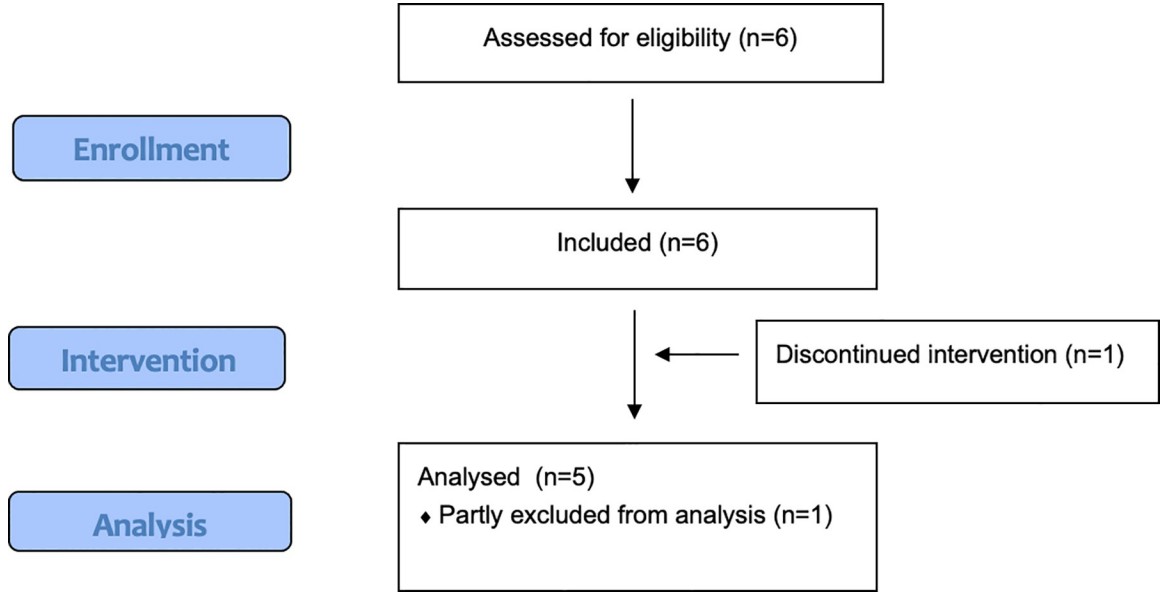

**Fig 1. Study flow chart.**

## Intervention

Participants were individually trained for 10–15 minutes by SJS in how to use the sleep robot. The at-home intervention lasted for 21 days. Participants were given the instruction to actively utilize the sleep robot while in bed, both before falling asleep and in case they woke up during the night. They were advised to place the robot against their abdomen and maintain contact with it. The robot's size and weight is 355 x 203 x 127 mm (14 x 8 x 5 in) and 1.9kg (4 lbs), respectively. The program named "sleeping" was used, which entailed 30 minutes of deep breathing with 1:2 inhalation and exhalation. Participants were free to use the robot for a shorter or a longer period of time, but were encouraged to use the robot each evening/night of the intervention phase. The robot's "breathing" could be manually changed via a control panel. The intervention is comparable with the first two peer-reviewed studies on the Somnox sleep robot [30, 38]. No incentives were given to increase compliance to the intervention.

## Outcome measures

**Sleep.** Daily subjective measurements of sleep onset latency (SOL), wake after sleep onset (WASO), total sleep time (TST), and sleep efficiency (SE) were obtained with the Consensus Sleep Diary for six consecutive weeks (two baseline weeks, three intervention weeks, and one post-intervention week) [39]. Questions about adherence and prescribed sleep medication use were included in the sleep diary. SOL, WASO, TST, and SE were also obtained objectively with wrist actigraphy (ActiGraph Link GT9X) for two weeks of the study: week 1 (baseline) and week 3 (intervention) [40]. The main outcome measure was WASO for Participant 1 and SOL for Participants 2–6 –their most salient insomnia symptom according to the ISI.

Insomnia Severity Index (ISI) is a well evaluated scale for insomnia symptoms and treatment effects. The questionnaire includes seven items with a total score range of 0–28 (Cronbach $\alpha = 0.74$) [34]. The cut-off of 11 (mild insomnia) was used in the current study. The established cut-offs for clinical improvements are -4.7 for a slight improvement, -8.4 for a moderate improvement, and -9.9 for a marked improvement [41]. Participants completed the ISI at four timepoints: pre-intervention, mid-intervention, post-intervention, and at 1-month follow-up.

**Somatic arousal.** Sleep-related arousal was measured with the somatic part of the Pre-Sleep Arousal Scale (PSAS) [42]. The somatic scale includes 8 items, with a total score range of 8–40 (Cronbach $\alpha = 0.72$) [43]. The cut-off of 10 was used in the current study to indicate hyperarousal, in line with Jansson-Fröjmark and Norell-Clarke [43]. Participants completed the PSAS at three timepoints: pre-intervention, post-intervention, and at 1-month follow-up.

**Emotional distress.** Symptoms of anxiety and depression were measured with the Hospital Anxiety and Depression Scale (HADS) [44]. The scale consists of two subscales (anxiety and depression), 14 items in total (7 in each scale) with a total score range of 0–21 (Cronbach $\alpha = 0.68$–0.93 for the anxiety scale, and Cronbach $\alpha = 0.67$–0.90 for the depression scale) [45]. Higher values mean more anxiety and depression symptoms, and a change of 1.5 points is considered the minimal important difference [46]. Participants completed the HADS at three timepoints: pre-intervention, post-intervention, and at the 1-month follow-up.

**ADHD.** The Adult ADHD Self-Report Scale (ASRS) was used to obtain self-reported ADHD symptoms (Swedish version ASRS-v1.1). The questionnaire consists of two subscales (inattention and hyperactivity/impulsivity) and 18 items in total (9 in each scale) with a total score range of 0–72 (Cronbach $\alpha = 0.88$) [47]. Higher values mean more ADHD symptoms. It is considered likely that the respondent meets the diagnostic criteria with a result of 17 or more in either subscale [48]. Participants completed the ASRS at three timepoints: pre-intervention, post-intervention, and at the 1-month follow-up.

## Quantitative analysis

Among other questions, participants reported sleep onset latency (SOL), wake after sleep onset (WASO), and total sleep time (TST) on a daily basis in the sleep diary. Sleep efficiency (SE: percentage of time spent asleep while in bed) was also calculated by dividing minutes spent asleep by the total amount of time in bed (in minutes). SOL, WASO, TST, and SE were also objectively measured with wrist actigraphy for two weeks of the study (1 and 3). The sleep diary and actigraphy scores were computed into weekly average scores. Study coordinator SJS was the only one who had access to information that could identify participants during and after data collection.

Data were analyzed by visual analysis [49]. To simplify the analysis, median lines of the three phases of the study (baseline, intervention, and post-intervention) were added to the graphs. The median is preferable to the mean with data distributions that fluctuate (as were the cases here). Additionally, the percentages of non-overlapping data (PAND) were calculated to compare the baseline and intervention phases. Parker and Vannest (p. 360) have defined PAND as "the smallest number of datapoints from either phase whose removal would eliminate all data overlap between two phases" [50]. Here, <50% overlap was considered an ineffective treatment, 50–70% an uncertain effect, 70–90% an effective treatment, and >90% a very effective treatment [51]. The computations were conducted in Microsoft Excel.

## Qualitative analysis

The interviews were audio-recorded and transcribed verbatim (see S1 File, Interview guide in the supporting information). The analytical approach was thematic analysis according to Braun and Clarke [52]. The analysis entailed six phases: (1) Reading and re-reading the interviews to familiarize with the material, (2) analyzing the material inductively for the core topics, and coding the topics, (3) relating codes into preliminary themes, (4) reviewing the themes, (5) naming and defining the themes, and finally, (6) selecting the extracts and producing the report [53]. The first author SJS coded the material. The themes were discussed with MT and ANC.

## Results

### Participant characteristics

Four participants were females and two were males (see Table 1). Two participants were in their twenties, one in his thirties, one in her forties, one in her fifties, and one in his sixties (age spans are provided instead of exact ages to anonymize participants as much as possible). One participant had graduated from elementary school, three participants from high school, and two participants had a university/college degree. Two participants were students, three had permanent employment, and one had retired. Three participants were living with a partner, the rest were singles. Two participants had children in the household, the rest had not. All were born in Sweden except for one. Four were using prescribed sleep medications, the other two were not. Participant 1 reported to have had insomnia symptoms for 10 years, Participant 2 and 5 for 20 years, Participant 3 for 8 years, Participant 4 for 15 years, and Participant 6 for 5 years.

### Outcome measures

**Sleep measures.** Fig 2 graphically depicts the wake after sleep onset (WASO) result for Participant 1, whereas Figs 3–6 show the sleep onset latency (SOL) results for Participants 2–5, respectively. Participant 6 did not complete the intervention due to life circumstances, which

**Table 1. Baseline characteristics of the participants.**

| | 1 | 2 | 3 | 4 | 5 | 6 |
|---|---|---|---|---|---|---|
| **Age span** | 50–59 | 60–69 | 20–29 | 30–39 | 40–49 | 20–29 |
| **Gender** | Female | Male | Female | Male | Female | Female |
| **Highest education** | University/college | High school | Elementary | High school | University/ college | High school |
| **Employment** | Permanent | Retired | Student | Permanent | Permanent | Student |
| **Marital status** | Single | Married/cohabitant | Single | Married/ cohabitant | Single | Married/ cohabitant |
| **Children in household** | No | Yes | No | No | Yes | No |
| **Born in Sweden** | Yes | No | Yes | Yes | Yes | Yes |
| **ADHD presentation** | Combined | Combined | Combined | Combined | Combined | Combined |
| **ADHD medication** | No | No | No | No | No | Yes |
| **Insomnia subtype** | Mid | Initial | Initial | Initial | Initial | Initial |
| **Insomnia medication** | No | Yes | Yes | No | Yes | Yes |
| **Insomnia duration years** | 10 | 20 | 8 | 15 | 20 | 5 |
| **ISI** | 20 | 18 | 22 | 20 | 17 | 11 |
| **PSAS** | 27 | 14 | 15 | 20 | 19 | 11 |
| **HADS–anxiety** | 16 | 8 | 11 | 8 | 9 | 8 |
| **HADS–depression** | 12 | 3 | 4 | 11 | 9 | 4 |
| **ASRS–inattention** | 18 | 32 | 20 | 20 | 30 | 27 |
| **ASRS–hyper/impulsivity** | 31 | 24 | 26 | 19 | 22 | 18 |

is why the sleep diary data is lacking. Data fluctuate in all three phases (baseline, intervention, post-intervention) of the study for all participants, making it hard to interpret data visually. Moreover, the actigraphy showed "0 minutes" for SOL on all days but one for one participant, that is, likely underreporting sleep onset latency.

With median lines of the three study stages added to the graphs, it is possible to see that WASO decreased for Participant 1 in the intervention phase. However, the percentage of non-overlapping data (PAND) was 29.03, representing an ineffective treatment. SOL seemed to decrease for Participant 2 in the intervention phase (and further decrease post-intervention), but the PAND was very low (8.57), denoting an ineffective treatment. SOL remained the same

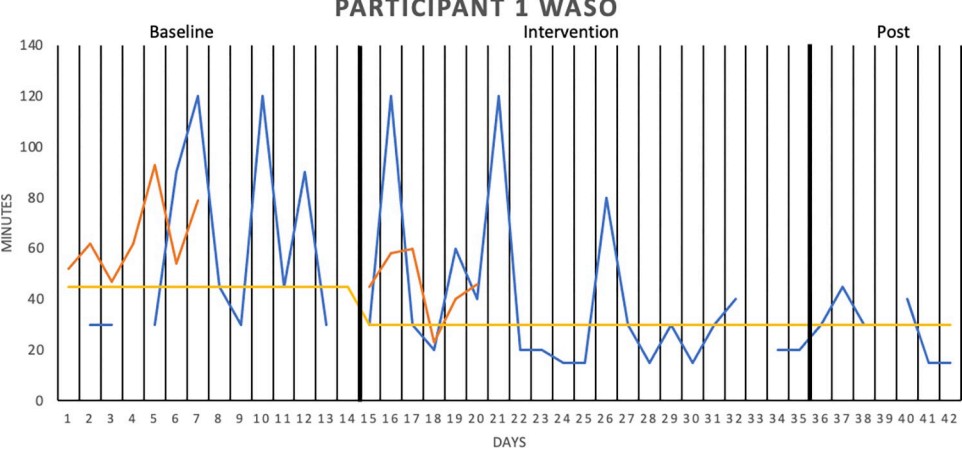

**Fig 2. Wake After Sleep Onset (WASO), in minutes for Participant 1.** WASO was measured over two baseline weeks, three intervention weeks, and one week post-intervention. The sleep diary is represented by the blue lines, the actigraphy by the orange lines, and the median of each phase by the yellow lines.

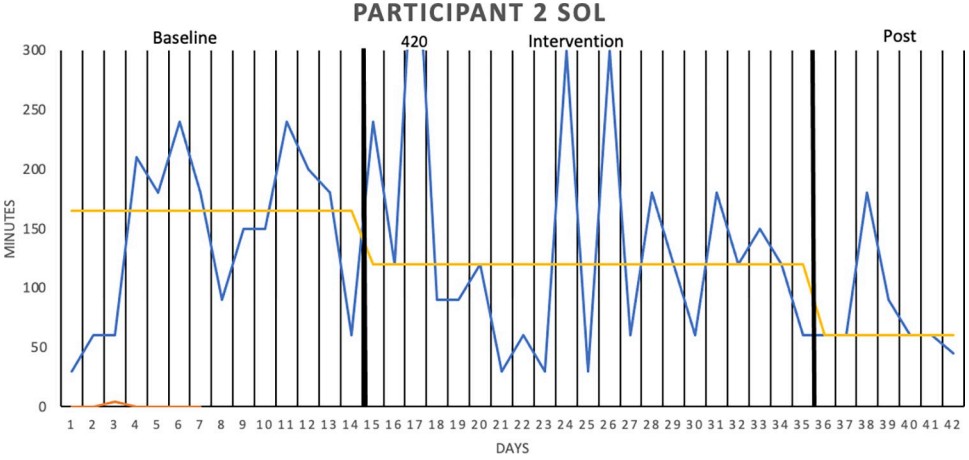

**Fig 3. Sleep onset latency (SOL), in minutes for Participant 2.** SOL was measured over two baseline weeks, three intervention weeks, and one week post-intervention. The sleep diary is represented by the blue lines, the actigraphy by the orange lines, and the median of each phase by the yellow lines.

as during baseline for Participants 3 (PAND = 23.53), 4 (PAND = 11.76) and 5 (PAND = 0). There was no support for the robot intervention being an effective treatment when it came to the other sleep diary variables either (see Table 2).

Regarding the level of insomnia symptoms as measured by the Insomnia Severity Index (ISI), the scores decreased for Participants 1–5 and remained the same for Participant 6 from pre- to mid-intervention. Participants 2, 3, and 5 reported further reductions post-intervention, whereas Participants 1 and 4 reported more insomnia symptoms post-intervention compared with mid-intervention. In terms of clinical relevance, Participant 2's change from pre- to post-intervention is considered a marked improvement, Participant 5's change a moderate improvement, and Participant 3's change a slight improvement [41]. From post-intervention to 1-month follow-up, Participants 1, 3, and 4 had reduced their ISI scores, whereas Participants 2 and 5 reported higher ISI scores. At the 1-month follow-up, all five participants who had completed the intervention experienced fewer insomnia symptoms compared with baseline (between -3 to -10 on the ISI). See Fig 7 for the ISI results presented graphically.

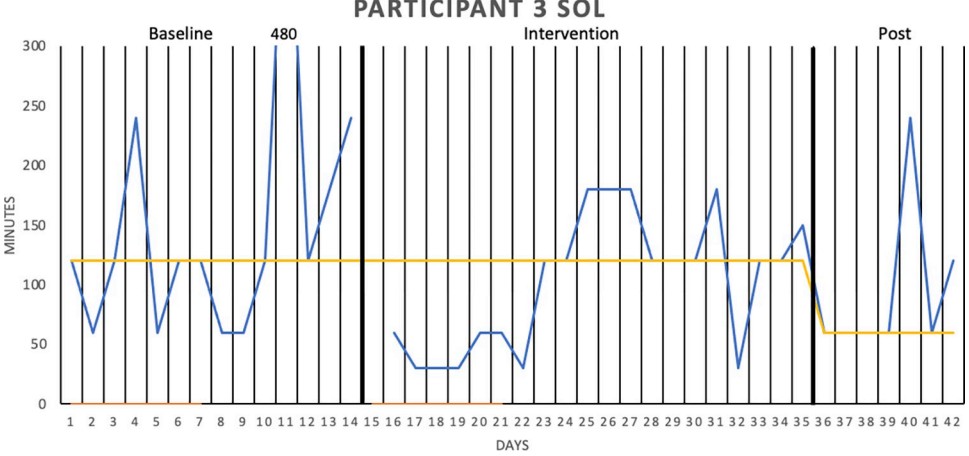

**Fig 4. Sleep onset latency (SOL), in minutes for Participant 3.** SOL was measured over two baseline weeks, three intervention weeks, and one week post-intervention. The sleep diary is represented by the blue lines, the actigraphy by the orange lines, and the median of each phase by the yellow lines.

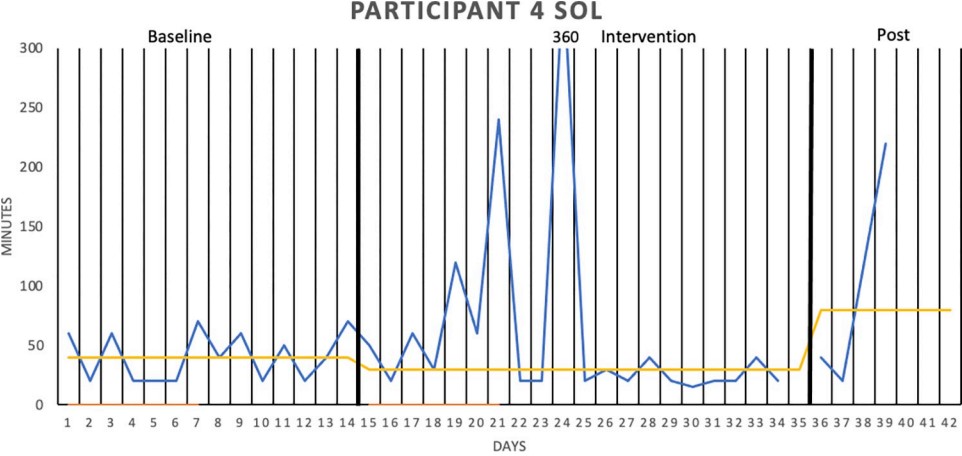

**Fig 5. Sleep onset latency (SOL), in minutes for Participant 4.** SOL was measured over two baseline weeks, three intervention weeks, and one week post-intervention. The sleep diary is represented by the blue lines, the actigraphy by the orange lines, and the median of each phase by the yellow lines.

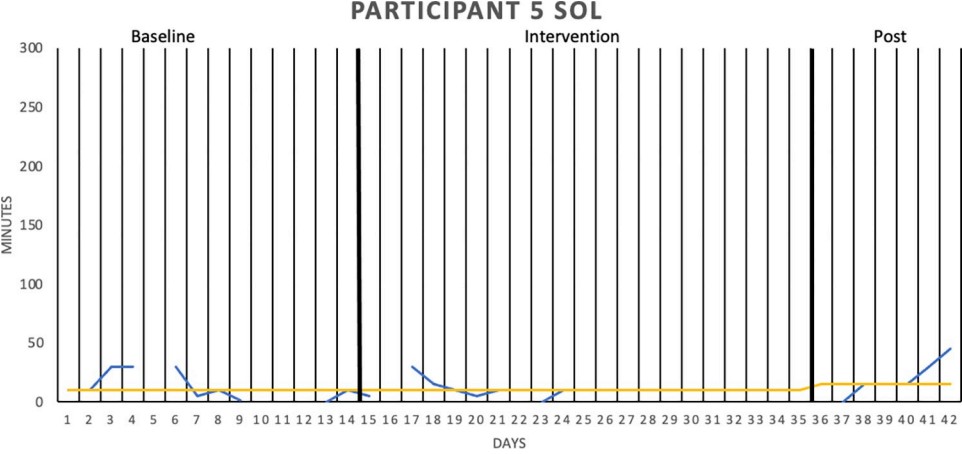

**Fig 6. Sleep onset latency (SOL), in minutes for Participant 5.** SOL was measured over two baseline weeks, three intervention weeks, and one week post-intervention. The sleep diary is represented by the blue lines, the actigraphy by the orange lines, and the median of each phase by the yellow lines.

**Table 2. The percentage of all non-overlapping data (PAND) for sleep onset latency (SOL), wake time after sleep onset (WASO), total sleep time (TST) and sleep efficiency (SE).** <50% overlap is considered an ineffective treatment, 50–70% an uncertain effect, 70–90% an effective treatment, and >90% a very effective treatment.

| Participant | SOL | WASO | TST | SE |
|---|---|---|---|---|
| 1 | 5.88 | 29.03 | 5.88 | 8.82 |
| 2 | 8.57 | 11.11 | 17.14 | 8.57 |
| 3 | 23.53 | - | 14.29 | 8.57 |
| 4 | 11.76 | 3.13 | 5.88 | 5.88 |
| 5 | 0 | - | 8.57 | 2.86 |

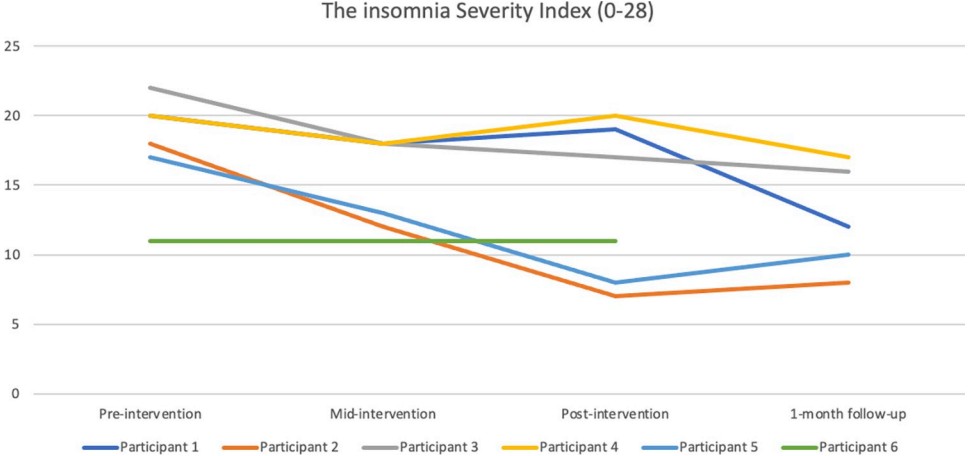

**Fig 7. Pre-intervention, mid-intervention, post-intervention, and 1-month follow-up assessments with The Insomnia Severity Index.**

**Somatic arousal.** According to the Pre-Sleep Arousal Scale (PSAS), Participants 1, 3, 4, and 5 reported reduced somatic arousal post-intervention compared with baseline, with further reductions for Participants 1 and 4 at 1-month follow-up. Participants 2 and 6 experienced slightly increased arousal post-intervention compared with baseline, and that remained on the same level for Participant 2 at 1-month follow-up. All participants scored above the cut-off of 10 post-intervention. Fig 8 depicts the PSAS results graphically.

**Emotional distress.** Regarding the Hospital Anxiety and Depression Scale–anxiety (see Fig 9), the only remarkable change was for Participant 5 from pre- to post-intervention with a large increase of anxiety symptoms (from 9 to 15). The score went down to 12 at 1-month follow up. Participant 3 had also increased anxiety post-intervention compared with baseline (from 11–13), representing a slight clinical important change for the worse. Regarding the depression scale (see Fig 10), Participant 5's symptoms had increased notably here as well from pre- to post-intervention. To a lesser degree, as had the symptom levels of Participant 1 and 3, and the changes are regarded as slightly important changes.

**ADHD.** Regarding the Adult ADHD Self-Report Scale–inattention (see Fig 11), Participants 1 and 5 reported reduced symptoms from baseline to post-intervention (below the

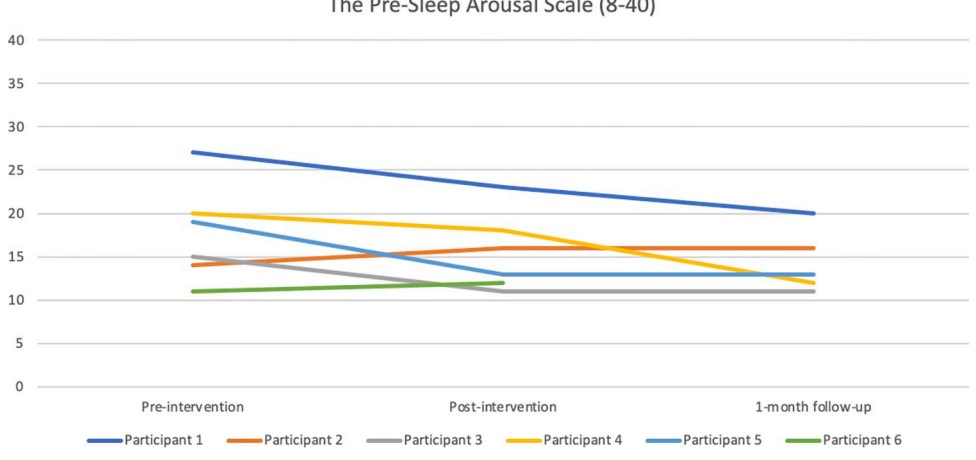

**Fig 8. Pre-, post-, and 1-month follow-up assessments with the Pre-Sleep Arousal Scale (PSAS).**

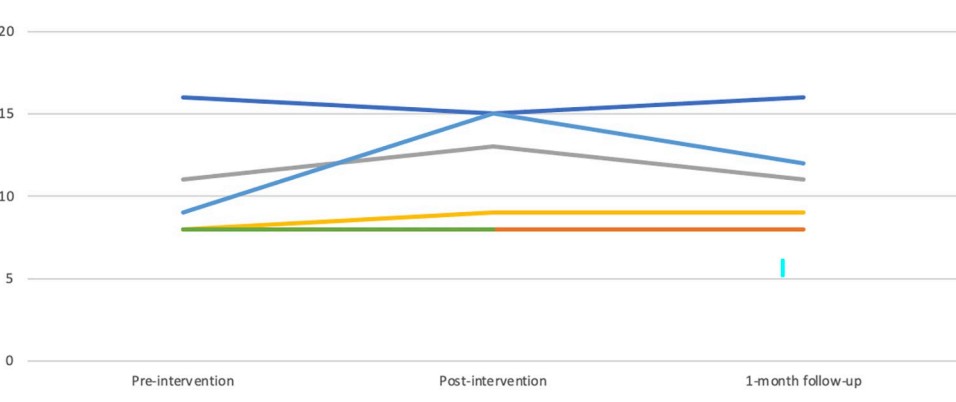

**Fig 9. Pre-, post-, and 1-month follow-up assessments with the Hospital Anxiety and Depression Scale—anxiety scale.**

threshold of 17 post-intervention for Participant 1), and a further reduction at 1-month follow up. Participant 6 also reported a somewhat reduced symptom burden post-intervention compared with baseline. Participant 2 remained at the same high level of symptoms throughout the study. Participants 3 and 4 reported more symptoms of inattention post-intervention compared with baseline and the follow-up data. Fig 12 shows the hyperactivity/impulsivity scale data. Participant 1 had nearly the same level of symptoms post-intervention compared with baseline, but reported fewer symptoms at 1-month follow-up. Participants 2, 4 and 6 scored more or less the same on all measurements. Participant 3's symptoms were fewer post-intervention compared with baseline, and remained so at 1-month follow-up, whereas Participant 5 reported more symptoms of hyperactivity/impulsivity post-intervention compared with baseline and 1-month follow-up.

## Adherence

Participant 1 used the robot 21 out of 21 nights (100%), for 60 minutes each night. Participant 2 used the robot 20 days (95.24%), for 120 minutes each night. Participant 3 used the robot

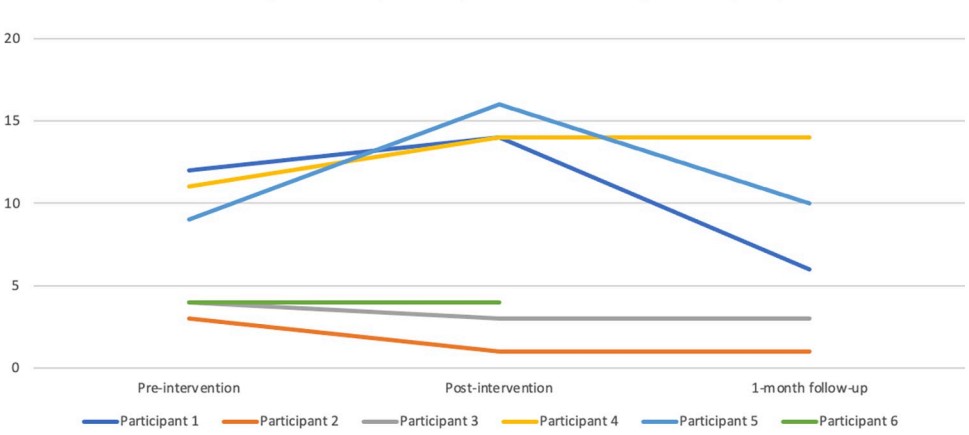

**Fig 10. Pre-, post-, and 1-month follow-up assessments with the Hospital Anxiety and Depression Scale—depression scale.**

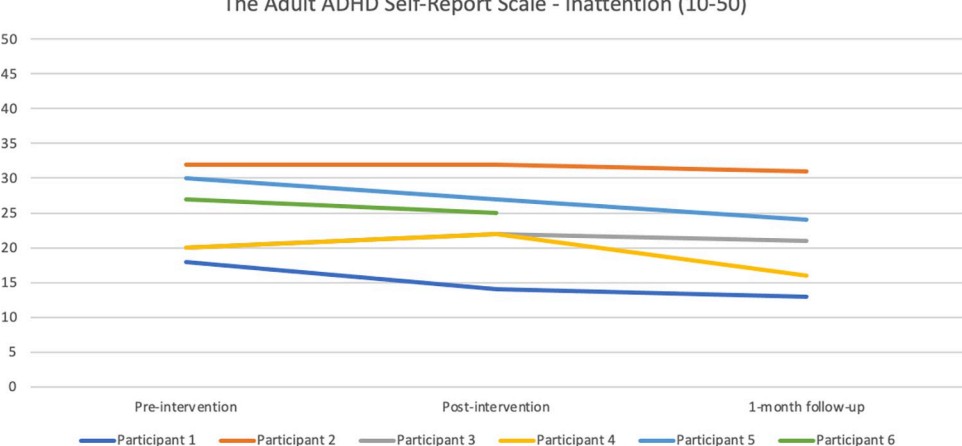

**Fig 11. Pre-, post-, and 1-month follow-up assessments with the Adult ADHD Self-Report Scale (ASRS)—inattention scale.**

every night (100%) for an average of 54:29 minutes each night. Participant 4 used the robot 3 out of 21 nights (14.29%); twice the first week and once the second week, for an average of 30 minutes each night. Participant 5 used the robot 10 out of 21 nights (47.62%): 4 nights the first week, 3 nights the second week, and 3 nights the third week, for an average of 27 minutes each night. Participant 6 used the robot 5 out of 21 days (23.81%), all during the first week of the intervention, for 60 minutes each night.

## The interview

The interview contained open-ended questions about participants' experiences with the intervention. The interviews lasted between 16:24 and 51:48 minutes and were conducted by the first author (SJS). Participants revealed both positive and negative experiences with the robot intervention. The thematic analysis resulted in 69 codes and three themes: (1) A pleasant companion, (2) Too much/not enough, and (3) A new routine. On the positive side, the robot was

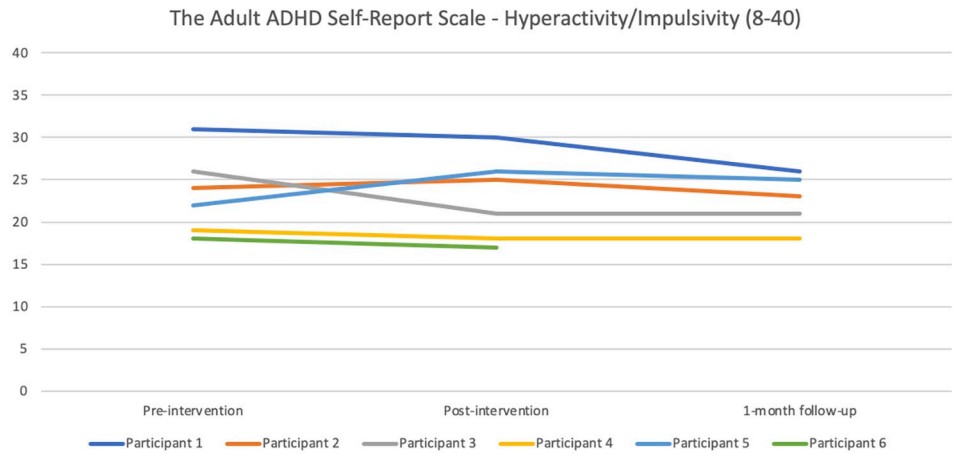

**Fig 12. Pre-, post-, and 1-month follow-up assessments with the Adult ADHD Self-Report Scale (ASRS)—hyperactivity/impulsivity scale.**

referred to as both pleasant and a good companion–even a substitute for human contact for those living by themselves. On the other hand, the robot did not meet the expectations of all participants, and was experienced as too big or too small, too noisy or too quiet, too hard or too soft etc. It was also a challenge for participants to start and maintain a new routine of using the robot every night, and to have to log their sleep over an extended period of time.

**A pleasant companion.**   The first theme was about the positive experiences that participants had with the robot in terms of sensations and companionship. The robot was described as "nice" and "relaxing" by most participants, and they liked its shape. Participants shared that they would try anything that might alleviate their sleeplessness. They mentioned that it took some time to get used to the robot, but that the transition into becoming comfortable with it was seamless.

Participant 1 felt that she calmed herself down faster together with the robot. She said that she had just as many middle-of-the-night awakenings as before, but that it was easier to fall back asleep when she used the robot, and for that reason she thought she had increased her total sleep time. She also stated that she missed the robot now that the intervention was over. Participant 2 also believed that he had had more total sleep time during the intervention, but was not sure whether it had anything to do with the robot. Participant 6 spoke of reduced arousal when she used the robot, and that it made it easier for her to reach the first stage of sleep:

> I think the robot made it easier to reach the preliminary stage of sleep, when it's like "okay, I'm about to fall asleep"–that felt nice. That's exactly what I struggle with, to reach that place.

Participants described the robot as "cute," "almost like a human" and as "a buddy." Participant 1 explained that she found it hard to sleep by herself and that the robot functioned as an alternative to human breathing and contact now that she was single:

> Before the intervention, when I shared a bed with my ex-partner, I would lie behind him and feel his breathing, which had a similar effect. So, it worked well on me. The robot calmed me down. . . In fact, it was a great substitute now that I sleep by myself.

She also thought that the robot helped her feel more secure:

> When I wake up at night, I sort of have this anxiety issue–I think a lot. It's awful to be awake in the small hours. I think the robot's been a security, like having another person there, it helped me feel safe.

Participants compared the intervention to mindfulness exercises. Participant 2 claimed that he "does not have the imagination for mindfulness," and that the robot might be a good alternative for those who struggle with exercises like *body scanning*, that is, with noticing sensations in their bodies:

> It's almost the same thing, you get the same feeling. But with mindfulness exercises, you have to do it yourself, you have to "go to your feet, then go higher". I'm not able to do that. I've tried it. . . I don't know why but I cannot picture it in my head: "Move to my feet, then to my legs," you know how it goes. It's not my thing.

He summarized his experience with the robot with the following words:

So, it was nice, but it wasn't the miracle that I hoped for.

**Too much/not enough.**   The second theme was about the negative experiences that participants had with the robot due to its form and functions. The theme also contains disclosures about the robot being more stressful than helpful and about not ever getting used to it. Participants complained that the 30-minute program was too short, and that the robot woke them up when it switched off. They also thought the battery time was below all expectations, and that it affected how much they used the robot during middle-of-the-night awakenings.

Participant 3 was one of the participants who experienced the robot as more stressful than helpful:

So, it stressed me out when I heard that it stopped, instead of helping me. . . I would start it again once more, and if I didn't fall asleep during the second round I would just give up. . . It's a shame that the robot didn't work for me, it would have been nice to just feel like, heck, that something worked.

Participants stated that the robot was either too soft or too hard, too big or too small. The robot was described as awkward, having to be moved back and forth in bed. Participant 2 thought that the robot's "respiration" was too slow/soft:

I liked it, you know, it felt good, but it wasn't loud enough.

Others thought that the robot was too loud. They were disturbed by the sounds the robot made in general, and specifically a mechanic "clicking" sound between each "breath." Participant 4 disclosed that even his partner had been bothered by the sounds of the robot, to the point where he thought he would have used the robot more often had it not been for that. He also complained that the robot invaded their space in bed. He suggested a combination of the robot and a weighted blanket, like the one he had used in the past, as the blanket would keep the robot in its place and muffle its sounds. At the same time, he did not really think any alterations of the intervention would make a difference:

I have a hard time falling asleep no matter what, and it's also the fact that when I sleep for eight or ten hours I'm as tired as when I've only slept for five hours, so, I don't think it would've made any difference. . .I have no idea why. It might be due to the ADHD in between.

Participant 5 was especially annoyed by the robot one night when she did not really feel the need for assistance:

When I did use it, it was quite nice, except for one evening when I got annoyed. But that was because I was so damn tired that I pretty much fell asleep the moment I laid my head down, and then I went like "oh no, the noise," and then it had the opposite effect.

She also made the point that the motivation to use the robot was much higher the nights when she actually slept poorly.

**A new routine.**   The third theme was about the intervention entailing a new routine for the participants, and how they struggled with both initiating and maintaining the routine. Participants stated that they would have wanted reminders on their phone (except for one

participant who thought she had too many reminders already) or another type of regular follow-up throughout the study. Participant 5, who often forgot to charge the robot in time for the night, dwelled on how this aspect might be especially challenging for patients with ADHD:

> That is a downside, even if it's just a charger it's an extra thing to do, it takes some time to make it a routine. . .I think the problem for us with ADHD is to keep our routines. That is what we struggle with, you know. And then there's so much happening in your life that can alter your routines. So, if you've succeeded with making good routines, something will happen that tear them down. . . I think that it has a lot to do with me having had an untreated ADHD for so long, I haven't had the correct treatment. That's important too. Ever since I was a child, nobody has helped me with establishing functioning routines. Now that I think about it, that can be part of the reason for my sleep problem.

Participants commented on how they experienced logging their sleep in the sleep diary. Participant 6 stated that it would have been better with a digital diary in her phone, as it was hard to remember to fill out the paper diary. She also thought that she probably would have had a higher adherence to the intervention had it not been for the sleep diary. Some participants described the sleep diary as interesting and spoke about the insight it gave them into their own sleep patterns, whereas others thought that they had a pretty clear picture of their sleep prior to the study.

## Discussion

The current mixed-method study is the first to explore individual effects and experiences with a three-week sleep robot intervention in adults with comorbid ADHD and insomnia. The main outcome measures were wake after sleep onset for Participant 1, and sleep onset latency for Participants 2–6 –measured both subjectively with a sleep diary and objectively with wrist actigraphy–as these were the participants' most prominent insomnia symptoms. The sleep diary and actigraphy data did not support any effects from the robot intervention, and data from the questionnaires were mixed. Analysis of the qualitative data resulted in themes that assist our understanding of the initial results. As Wiart, Rosychuk and Wright (p. 6) have stated, "Knowing *why* interventions do or do not work is as important as knowledge of effectiveness if interventions are to be successfully transferred into 'real world' clinical settings." The qualitative findings will be discussed in conjunction with quantitative findings [53].

Regarding levels of insomnia symptoms according to the Insomnia Severity Index (ISI), Participant 2 reported a marked improvement from pre- to post-intervention (-11), whereas Participant 5 reported a moderate improvement (-9), and Participant 3 a slight improvement (-5) [41]. Moreover, Participant 2 and 5 scored below the cut-off post-intervention. The rest of participants scored precisely or nearly the same on the ISI post-intervention as they did at baseline. Participants 2 and 3 used the robot all nights of the intervention except for one night for Participant 2, whereas Participant 5 used the robot almost 50% of the nights, indicating that the results might reflect a dose-response relationship for responders. The ISI results were interesting, as they were not always in line with the sleep diary variables, nor with what participants disclosed in the interviews. For instance, Participant 1 spoke fondly of the robot as something that made her feel more safe and secure. She explicitly stated that she believed she had as many middle-of-the-night awakenings during the intervention as she did prior to it, but that she fell asleep faster during the intervention, and thus she believed she had increased her total sleep time. This is intuitively not in line with the no-change on the ISI, although it might be explained by the fact that the questionnaire reflects general insomnia symptoms over a week

(more than estimated sleep times) and that the diagnostic criteria of insomnia do not include objective short sleep time. Conversely, Participant 2 described the robot intervention as "nice," but not the miracle he had hoped for, and yet he reported a marked improvement on the ISI.

All participants had a high level of somatic arousal according to the Pre-Sleep Arousal Scale (PSAS) at baseline with scores ranging between 11–27, where a score of 10 represents hyper-arousal in several studies [28, 43], including the current study. Participants 1, 3, 4 and 5 reported reduced arousal from pre- to post-intervention (ranging from -2 to -6), whereas the other two participants reported increased arousal (+1 and +2 from pre- to post-intervention). All participants did, however, score above cut-off on the PSAS post-intervention. Relaxation has been found to be pleasant in itself, which is why participants may describe the robot as nice and pleasant and at the same time not helpful regarding their sleep-disturbing arousal. Moreover, the study did not include an objective measure of participants' breathing during the intervention, why we do not know whether the lack of effects were due to the robot not affecting participants breathing, or whether potentially changed breathing did not affect participants' sleep. Participants compared the robot to mindfulness exercises, and pointed out that the robot might be an alternative for those who find it hard to execute such exercises without support.

Regarding the Adult ADHD Self-Report Scale (ASRS), Participants 1 and 5 had fewer symptoms of inattention (but not of hyperactivity/impulsivity) post-intervention (from 18 to 14 for Participant 1, i.e., below the cut-off) compared with baseline. Relating the results to the results on the other symptom questionnaires, Participant 5 reported a marked improvement on the ISI from pre- and post-intervention, whereas Participant 1 remained on the same level of insomnia symptoms, i.e., not similar trends. Both participants reported a reduction on the PSAS of -4 points from pre- to post-intervention. Participant 2 reported basically the same number of symptoms on both ASRS scales at the two time points. Participants 3 and 4 reported slightly more symptoms of inattention from pre- to post-intervention (+2), which were not in line with their beneficial changes on the ISI and the PSAS. Participant 3 did however report fewer symptoms of hyperactivity/impulsivity post-intervention compared with baseline. As the relationship between insomnia and ADHD is a complex one, and as the sleep robot intervention does not target ADHD per se, the minimal changes and unclear pattern is perhaps to be expected.

The study has several limitations. The first limitation is that it only includes six participants. Thus, the study can only speak on individual effects of the sleep robot intervention, that is, it has limited external validity. Three participants had a 100% or close to 100% adherence in terms of using the robot every night of the intervention phase of the study (Participants 1–3). The other three used to robot less than 50% of the nights, which might be a reason for the limited effects. As one participant stated, the motivation to use the robot was substantially higher the nights she actually had a hard time falling asleep. Since insomnia symptoms fluctuate (the diagnostic criteria only demand that the sleep problem occurs three times a week), this is perhaps also in line with how a person would use the robot after buying one. Another limitation was that data fluctuated a great deal on all sleep outcome measures for all participants throughout the study, lowering the internal validity of the study and complicating interpretation.

In a repeated ABA single-case design, all cases are considered their own experiment, why it is positive for the internal validity that we have six repeated cases. The study can be said to have decent quantitative reliability and validity as psychometrically sound instruments were used. Regarding qualitative validity, the study followed the recommended strategies listed in Creswell and Plano Clark (p. 217), including triangulation of data from several sources and presenting disconfirming evidence [31]. As subjective and objective sleep measures do not always correspond, a strength with the current study is the collection of both. Other strengths

of the study are the exhaustive screening of participants, and the collecting of adherence data. A further strength is the calculation of effect sizes through the percentage of all non-overlapping data (PAND) in addition to visual inspection of data, as the latter is associated with subjectivity and inconsistency [54]. The study can also be said to have a high ecological validity due to the at home-intervention, in line with how it would be if the participants bought the robot themselves.

Since "absence of evidence is not evidence of absence" [55], the authors recommend more studies to be conducted on the sleep robot intervention in adults with ADHD and insomnia. However, before more studies are conducted, the intervention should be adjusted according to the current results. We recommend collection of sleep diary and actigraphy data pre- and post-intervention only, as to lessen the burden of participation and ensure that the participants can have their sole focus on the robot during the intervention. Level of emotional distress should, on the other hand, be examined more often throughout the intervention. Since participants disclosed cognitive arousal in the interviews, a questionnaire on cognitive arousal should be included. Objective measurements of participants' breathing would also yield important information. As participants compared the intervention to mindfulness exercises, another option is to include a mindfulness outcome measure. Digital collection of data is preferable, as are digital reminders to the participants.

Future studies could consider including psychoeducation on sleep hygiene, stimulus control and sleep restriction in the intervention, as the robot itself might not affect circadian rhythm and sleep pressure. Another idea for the future is to compare effects of the sleep robot with effects of weighted blankets, which are commonly prescribed to people with ADHD in Sweden. Quantitative studies should also consider including interviews with at least a sample of the participants, that is, continue to explore participants' experiences with the robot to both broaden and deepen our knowledge about the intervention. To conclude, the effects of the sleep robot intervention were mixed. More studies must be conducted on a modified version of the intervention before it can be recommended to adults with ADHD and insomnia in clinical settings.

## Supporting information

**S1 Checklist. TREND statement checklist.**
(PDF)

**S1 File. Interview guide.**
(DOCX)

**S2 File. Study protocol English.**
(DOCX)

**S3 File. Study protocol Swedish.**
(PDF)

**S4 File. Adjusted study protocol English.**
(DOCX)

**S5 File. Adjusted study protocol Swedish.**
(DOCX)

## Acknowledgments

The authors are grateful to the participants for their time.

## Author Contributions

**Conceptualization:** Siri Jakobsson Støre, Maria Tillfors, Annika Norell-Clarke.

**Data curation:** Siri Jakobsson Støre, Charlotte Angelhoff.

**Formal analysis:** Siri Jakobsson Støre.

**Investigation:** Siri Jakobsson Støre, Charlotte Angelhoff.

**Methodology:** Siri Jakobsson Støre, Maria Tillfors, Annika Norell-Clarke.

**Project administration:** Siri Jakobsson Støre.

**Software:** Siri Jakobsson Støre.

**Supervision:** Maria Tillfors, Annika Norell-Clarke.

**Writing – original draft:** Siri Jakobsson Støre.

**Writing – review & editing:** Maria Tillfors, Charlotte Angelhoff, Annika Norell-Clarke.

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
