## [Decision Letter · Decision Letter 0]

19 Jun 2023

PONE-D-23-08448A robot intervention for adults with adhd and insomnia

– A mixed-method studyPLOS ONE

Dear Dr. Støre,

Thank you for submitting your manuscript to PLOS ONE. After careful consideration, we feel that it has merit but does not fully meet PLOS ONE’s publication criteria as it currently stands. Therefore, we invite you to submit a revised version of the manuscript that addresses the points raised during the review process.

We look forward to receiving your revised manuscript.

Kind regards,

Armando D'Agostino

Academic Editor

PLOS ONE

Journal Requirements:

Additional Editor Comments:

Although the intervention is of interest, the study is very limited in terms of participants and may only be presented as a proof-of-concept study. Please include this term in your title. Moreover, further detail should be given on the Somnox sleep robot (which the majority of readers will not have heard of before finding this paper). A Figure including the robot image and summary of characteristics may also be helpful.

Participants and Procedure

"The sample size of 6 was determined based on the feasibility nature of the study (i.e., new patient group) combined with the number of sleep robots at hand (a deviation from the original study

protocol were we wanted to recruit 20 participants)".

It is unclear why the study ended up with such a small number of participants: were the authors not aware of the available sleep robots when the study began? Did the ethics committee approve the presented limited-sample mixed-methods study or was a different trial presented?

Reviewers' comments:

Reviewer's Responses to Questions

**Comments to the Author**

1. Is the manuscript technically sound, and do the data support the conclusions?

Reviewer #1: Partly

2. Has the statistical analysis been performed appropriately and rigorously? 

Reviewer #1: I Don't Know

3. Have the authors made all data underlying the findings in their manuscript fully available?

Reviewer #1: Yes

4. Is the manuscript presented in an intelligible fashion and written in standard English?

Reviewer #1: Yes

5. Review Comments to the Author

Reviewer #1: Important note: This review pertains only to ‘statistical aspects’ of the study and so ‘clinical aspects’ [like medical importance, relevance of the study, ‘clinical significance and implication(s)’ of the whole study, etc.] are to be evaluated [should be assessed] separately/independently. Further please note that any ‘statistical review’ is generally done under the assumption that (such) study specific methodological [as well as execution] issues are perfectly taken care of by the investigator(s). This review is not an exception to that and so does not cover clinical aspects {however, seldom comments are made only if those issues are intimately / scientifically related & intermingle with ‘statistical aspects’ of the study}. Agreed that ‘statistical methods’ are used as just tools here, however, they are vital part of methodology [and so should be given due importance]. I look at the manuscript in/with statistical view point, other reviewer(s) look(s) at it with different angle so that in totality the review is very comprehensive. However, there should be efforts from authors side to improve (may be by taking clues from reviewer’s comments). Therefore, please do not limit the revision only (with respect) to comments made here.

COMMENTS: Although this manuscript is well drafted [and the study is excellent with respect to most of the aspects], I have few observations/concerns (different opinion) which are given below:

Firstly, I suggest a very minor correction. In the title [Title: “A robot intervention for adults with adhd and insomnia – A mixed-method study”] and subsequently at many more places [including at least five times in ‘Abstract’] ‘adhd’ is in small letters. It should be in capital letters ‘ADHD’ as it is a long-form/abbreviation which stands for ‘Attention Deficit & Hyperactivity Disorder’. Therefore, the required change is requested.

Since conclusion of this study [To conclude, the effects of the sleep robot intervention were mixed. More studies must be conducted on a modified version of the intervention before it can be recommended to adults with adhd and insomnia in clinical settings] is similar to those of quoted {both excellent and by the same team of scientists} studies (reference numbers 30 & 38 and thanks for giving DOI for both), I wish to know “why this study [though participant characteristics may be different] and moreover why on very small sample? Authors may have justification, but need to clarify.

Authors are likely to be aware of fact that: “Absence of evidence is not evidence of absence” [Altman DG, Bland JM. BMJ volume 311, 1995, p 485 (Reprinted: Australian Veterinary Journal 1996;74, 311)]. {Even when P-value is not significantly lower that is null hypothesis of no difference / no association is not rejected, (in short, result is not significant), that does not amount to evidence of absence i.e., it does not imply that there no difference / no association. It only implies that there is no (i.e., these samples do not provide) [say enough] evidence to prove (rather indicate with certain specified confidence level) the difference / association}. However, the is question is “HOW MUCH ETHICAL IT IS TO OFFER AN UNEFFECTIVE INTERVENTION TO WAIT-LISTED CONTROLS”? This question is not limited to your trial only and agreed that it may be the universal question. But how you dealt with ‘WAIT-LISTED CONTROLS’ in your trial is specific or pertaining to this trial and needs to be answered.

As you said “Considering the neurochemical and anatomical overlap in sleep, arousal and attention, it is possible that the robot might have greater effects in adults with comorbid adhd and insomnia” and I agree, however, the question of ‘small sample’ needs to be justified yet. You said “The sample size of 6 was determined based on the feasibility nature of the study” but then why title does/do not include/show that this is a feasibility study? It is also said that “The study was registered retrospectively due to the pilot nature of the study” but no mention of pilot nature of the study in ‘Abstract’. Considering variation in ‘Participant characteristics’ {ex. Age} and measurements (ex., The main outcome measure was WASO for Participant 1 and SOL for Participants 2-6) – tools or results/observations/recording - I sincerely of the opinion that it can be treated only as a pilot/feasibility study and needs clear mention in ‘Abstract’.

In ‘Intervention’ section it is stated that “Participants were free to use the robot for a shorter or a longer period of time”. Research methodology of ‘clinical trial’ can accept such point only (to some extent) if the trial is ‘pragmatic’. Except these minor points, the article is acceptable. However, mind you that as pointed out in ‘important note’ above “This review pertains only to ‘statistical aspects’ of the study and so ‘clinical aspects’ should be assessed separately/independently. ‘Minor Revision’ is recommended. Can be accepted only as a pilot/feasibility study after revision.

6. PLOS authors have the option to publish the peer review history of their article (what does this mean?). If published, this will include your full peer review and any attached files.

Reviewer #1: No

---

## [Author Response · Author response to Decision Letter 0]

29 Jun 2023

Journal Requirements:

Answer: Thank you very much for the opportunity to revise the manuscript. The manuscript has now been formatted according to the PLOS ONE templates.

Answer: A de-identified data set is now publicly available from the ISRCTN registry (ISRCTN11007746).

Answer: The caption for the supporting information file has been added at the end of the manuscript, and the in-text citation has been changed as well.

Additional Editor Comments:

Although the intervention is of interest, the study is very limited in terms of participants and may only be presented as a proof-of-concept study. Please include this term in your title. Moreover, further detail should be given on the Somnox sleep robot (which the majority of readers will not have heard of before finding this paper). A Figure including the robot image and summary of characteristics may also be helpful.

Answer: Thank you for your valuable comments. The term proof-of concept has been added to the title. 

The following sentences have been added about the Somnox sleep robot in the intervention section (page 7), as well as an image of the robot: “Participants were given the instruction to actively utilize the sleep robot while in bed, both before falling asleep and in case they woke up during the night. They were advised to place the robot against their abdomen and maintain contact with it. The robot’s size and weight is 355 x 203 x 127 mm (14 x 8 x 5 in) and 1.9kg (4 lbs), respectively.”

Participants and Procedure

"The sample size of 6 was determined based on the feasibility nature of the study (i.e., new patient group) combined with the number of sleep robots at hand (a deviation from the original study

protocol were we wanted to recruit 20 participants)".

It is unclear why the study ended up with such a small number of participants: were the authors not aware of the available sleep robots when the study began? Did the ethics committee approve the presented limited-sample mixed-methods study or was a different trial presented?

Answer: The sentence has been changed to “The sample size of 6 was determined based on the feasibility nature of the study (i.e., new patient group) due to the lack of clear beneficial effects in the preceding study”, which we believe makes more sense.

Reviewers' comments:

Review Comments to the Author

Reviewer #1: 

COMMENTS: Although this manuscript is well drafted [and the study is excellent with respect to most of the aspects], I have few observations/concerns (different opinion) which are given below:

Answer: Thank you for your positive comments on the manuscript. Here follows responses to your concerns.

Firstly, I suggest a very minor correction. In the title [Title: “A robot intervention for adults with adhd and insomnia – A mixed-method study”] and subsequently at many more places [including at least five times in ‘Abstract’] ‘adhd’ is in small letters. It should be in capital letters ‘ADHD’ as it is a long-form/abbreviation which stands for ‘Attention Deficit & Hyperactivity Disorder’. Therefore, the required change is requested.

Answer: We agree and have changed adhd to ADHD with capital letters throughout the manuscript.

Since conclusion of this study [To conclude, the effects of the sleep robot intervention were mixed. More studies must be conducted on a modified version of the intervention before it can be recommended to adults with adhd and insomnia in clinical settings] is similar to those of quoted {both excellent and by the same team of scientists} studies (reference numbers 30 & 38 and thanks for giving DOI for both), I wish to know “why this study [though participant characteristics may be different] and moreover why on very small sample? Authors may have justification, but need to clarify.

Answer: Thank you. The rationale for the small sample size has been changed according to a previous comment: “The sample size of 6 was determined based on the feasibility nature of the study (i.e., new patient group) due to the lack of clear beneficial effects in the preceding study.”

Authors are likely to be aware of fact that: “Absence of evidence is not evidence of absence” [Altman DG, Bland JM. BMJ volume 311, 1995, p 485 (Reprinted: Australian Veterinary Journal 1996;74, 311)]. {Even when P-value is not significantly lower that is null hypothesis of no difference / no association is not rejected, (in short, result is not significant), that does not amount to evidence of absence i.e., it does not imply that there no difference / no association. It only implies that there is no (i.e., these samples do not provide) [say enough] evidence to prove (rather indicate with certain specified confidence level) the difference / association}. However, the is question is “HOW MUCH ETHICAL IT IS TO OFFER AN UNEFFECTIVE INTERVENTION TO WAIT-LISTED CONTROLS”? This question is not limited to your trial only and agreed that it may be the universal question. But how you dealt with ‘WAIT-LISTED CONTROLS’ in your trial is specific or pertaining to this trial and needs to be answered.

Answer: Thank you, we agree and have added the following sentence in the discussion (p. 20): Since “absence of evidence is not evidence of absence, (54) the authors recommend more studies to be conducted on the sleep robot intervention in adults with ADHD and insomnia. However, before more studies are conducted, the intervention should be adjusted according to the current results.”

We do not have waitlist controls in the current study, why we have not dealt with the latter part of this comment.

As you said “Considering the neurochemical and anatomical overlap in sleep, arousal and attention, it is possible that the robot might have greater effects in adults with comorbid adhd and insomnia” and I agree, however, the question of ‘small sample’ needs to be justified yet. You said “The sample size of 6 was determined based on the feasibility nature of the study” but then why title does/do not include/show that this is a feasibility study? It is also said that “The study was registered retrospectively due to the pilot nature of the study” but no mention of pilot nature of the study in ‘Abstract’. Considering variation in ‘Participant characteristics’ {ex. Age} and measurements (ex., The main outcome measure was WASO for Participant 1 and SOL for Participants 2-6) – tools or results/observations/recording - I sincerely of the opinion that it can be treated only as a pilot/feasibility study and needs clear mention in ‘Abstract’.

Answer: Thank you for pinpointing the absence of pilot/feasibility terms in the title and the abstract. The title and abstract now includes the term proof-of-concept study to accommodate a previous comment.

In ‘Intervention’ section it is stated that “Participants were free to use the robot for a shorter or a longer period of time”. Research methodology of ‘clinical trial’ can accept such point only (to some extent) if the trial is ‘pragmatic’. Except these minor points, the article is acceptable. However, mind you that as pointed out in ‘important note’ above “This review pertains only to ‘statistical aspects’ of the study and so ‘clinical aspects’ should be assessed separately/independently. ‘Minor Revision’ is recommended. Can be accepted only as a pilot/feasibility study after revision.

Answer: Thank you very much. As mentioned above, the study is now framed as a proof-of-concept study.

---

## [Decision Letter · Decision Letter 1]

21 Aug 2023

A robot intervention for adults with adhd and insomnia

– A mixed-method proof-of-concept study

PONE-D-23-08448R1

Dear Dr. Støre,

We’re pleased to inform you that your manuscript has been judged scientifically suitable for publication and will be formally accepted for publication once it meets all outstanding technical requirements.

Kind regards,

Armando D'Agostino

Academic Editor

PLOS ONE

Additional Editor Comments (optional):

Reviewers' comments:

Reviewer's Responses to Questions

**Comments to the Author**

1. If the authors have adequately addressed your comments raised in a previous round of review and you feel that this manuscript is now acceptable for publication, you may indicate that here to bypass the “Comments to the Author” section, enter your conflict of interest statement in the “Confidential to Editor” section, and submit your "Accept" recommendation.

Reviewer #1: All comments have been addressed

2. Is the manuscript technically sound, and do the data support the conclusions?

Reviewer #1: (No Response)

3. Has the statistical analysis been performed appropriately and rigorously? 

Reviewer #1: (No Response)

4. Have the authors made all data underlying the findings in their manuscript fully available?

Reviewer #1: (No Response)

5. Is the manuscript presented in an intelligible fashion and written in standard English?

Reviewer #1: (No Response)

6. Review Comments to the Author

Reviewer #1: COMMENTS: All the comments are answered and positively attended [good that the study now is framed as a proof-of-concept study & mention in title]. I recommend the acceptance because the manuscript has now achieved the acceptable level in my opinion (as a proof-of-concept study).

7. PLOS authors have the option to publish the peer review history of their article (what does this mean?). If published, this will include your full peer review and any attached files.

Reviewer #1: **Yes: **Dr. Sanjeev Sarmukaddam

---

## [Editor Report · Acceptance letter]

24 Aug 2023

PONE-D-23-08448R1 

A robot intervention for adults with ADHD and insomnia
– A mixed-method proof-of-concept study 

Dear Dr. Støre:

I'm pleased to inform you that your manuscript has been deemed suitable for publication in PLOS ONE. Congratulations! Your manuscript is now with our production department. 

Kind regards, 

on behalf of

Dr. Armando D'Agostino 

Academic Editor

PLOS ONE